# Synthesis of Bio-Based Polyester from Microbial Lipidic Residue Intended for Biomedical Application

**DOI:** 10.3390/ijms24054419

**Published:** 2023-02-23

**Authors:** Ana P. Capêto, João Azevedo-Silva, Sérgio Sousa, Manuela Pintado, Ana S. Guimarães, Ana L. S. Oliveira

**Affiliations:** 1Centro de Biotecnologia e Química Fina (CBQF)-Laboratório Associado, Escola Superior de Biotecnologia, Universidade Católica Portuguesa, Rua Diogo de Botelho 1327, 4169-005 Porto, Portugal; 2CONSTRUCT, Faculdade de Engenharia do Porto (FEUP), Universidade do Porto, Rua Doutor Roberto Frias, 4200-465 Porto, Portugal

**Keywords:** microbial oil, bio-based polyester, biocompatible, hydrophilic, wound dressing, drug delivery

## Abstract

In the last decade, selectively tuned bio-based polyesters have been increasingly used for their clinical potential in several biomedical applications, such as tissue engineering, wound healing, and drug delivery. With a biomedical application in mind, a flexible polyester was produced by melt polycondensation using the microbial oil residue collected after the distillation of β-farnesene (FDR) produced industrially by genetically modified yeast, *Saccharomyces cerevisiae.* After characterization, the polyester exhibited elongation up to 150% and presented *T_g_* of −51.2 °C and *T_m_* of 169.8 °C. In vitro degradation revealed a mass loss of about 87% after storage in PBS solution for 11 weeks under accelerated conditions (40 °C, RH = 75%). The water contact angle revealed a hydrophilic character, and biocompatibility with skin cells was demonstrated. 3D and 2D scaffolds were produced by salt-leaching, and a controlled release study at 30 °C was performed with Rhodamine B base (RBB, 3D) and curcumin (CRC, 2D), showing a diffusion-controlled mechanism with about 29.3% of RBB released after 48 h and 50.4% of CRC after 7 h. This polymer offers a sustainable and eco-friendly alternative for the potential use of the controlled release of active principles for wound dressing applications.

## 1. Introduction

The transition to a higher degree of sustainability lies in the industrial development of bio-based and biodegradable plastics to counteract the absence of new extensive sources of conventional non-renewable crude oil, avoiding the related detrimental effect on climate change and environmental pollution [1,2].

In this context, resource circulation, a concept that combines circular economy and sustainable development goals, has been used to replace the conventional petroleum refinery with an economically viable biorefinery targeting the production of several bio-based materials, including plastics [3]. New platform chemicals [4,5] were created, based on carbon-rich precursors (nucleic acids, proteins, carbohydrates, lipids, etc.) derived from the second and third-generation expanded range of feedstocks, such as lignocellulosic residues (e.g., from agriculture, forest, and industrial by-products), and extracts obtained from wastes of industrial, and municipal origin [6,7]. 

Natural (e.g., alginate, cellulose, starch, etc.) and synthetic biodegradable polymers (e.g., polyesters, polyamides, etc.) are produced for high-value applications in the food, pharmaceutical, and biomedical fields through microbial agents, biopolymer blending, or/and chemical methods [8,9,10]. 

The most well-known and characterized bio-based polymers and copolymers [11,12,13,14] are synthesized from monomers, directly or indirectly obtained by fermentation of biomass, such as α-hydroxy acids (e.g., citric, lactic, acetic, malic, glycolic, and tartaric acids) [15]; alcohols and glycols (e.g., ethanol, butanediol, propanediol, glycerol, etc.); dicarboxylic acids (e.g., succinic, sebacic, itaconic, azelaic acids, etc.); and diamines [16,17,18].

Polylactide (PLA) and poly lactide-co-glycolide (PLGA) are good examples of biodegradable poly(α-hydroxy acids) aliphatic polyesters with extensive biomedical applications in disposable and support products (e.g., syringes, blood bags, sutures, bone plates, and sealants), hard and soft tissue engineering, surgical implants, reconstructive surgery, wound dressings, controlled drug delivery systems, among others [19,20]. These biopolymers allow precise control of their properties, replicability of results, and easy adaptation to industrial production [21,22]. Beyond the economic and technical restrictions to this kind of application, polymers need to meet other hard limits such as biocompatibility, mechanical/thermal performance, wettability, or stability [23,24]. 

Skin diseases are a serious burden in healthcare and chronic wound treatment, representing a global care cost varying from $13 to $15 billion annually [25]. Based on this, the use of novel technologies for the treatment of skin diseases and injuries is highly encouraged. The development of site-specific drug delivery represents an advantage because it is painless and noninvasive, while improving drug bioavailability, and it minimizes the systemic toxicity and drug exposure to non-desired sites [26].

Recent advances in green technological processes allowed for the development of new bio-based materials or composites to produce semi-permeable films, foams, hydroactive dressings, and hydrogels [27,28,29], designed to have a more active role in advanced cutaneous wound healing [30,31]. These smart materials provide a physical shield that avoids external bacterial contamination of the wound, allowing gas exchange, while maintaining proper moisture by absorbing wound exudates [32,33]. Additionally, these materials are suitable for drug delivery, controlling the release kinetics, and increasing the performance and effectiveness of the treatment [34,35,36,37,38].

Several natural synthetic polymers have been used for tissue engineering and wound healing applications since they combine properties such as biocompatibility and or biodegradability, and despite development strategies, performance and price are still hot topics, since to achieve the required properties, biopolymers have high production costs [39]. Within this context, the utilization of waste material or renewable resources may overcome this problem [40]. 

The aim of this work was to develop a flexible and biocompatible aliphatic copolyester from a biowaste, a microbial oil, to produce a porous scaffold for wound dressing, suitable for the controlled release of bioactive compounds. 

## 2. Results and Discussion

### 2.1. Synthesis and Characterization of FDR-Based Epoxide

The microbial oil residue collected after distillation of β-farnesene (FDR) industrial production by a fermentation process was characterized elsewhere [41] by the presence of hydrocarbons (43.29% wt. lipids), terpenes (e.g., farnesene and farnesol), a small percentage of fatty acids (0.33%), complex lipids such as triglycerides (4.29%), diglycerides (2.62%), among other compounds.

FDR has a biphasic nature with a certain amount of waxes esters (0.22%) that increase the viscosity of the derived polymeric materials due to the formation of crystals at room temperature [42]. The material’s viscosity is an important parameter that affects productivity and product quality, along with polymer processability, polymerization kinetics (mass and heat transfer coefficients, residence time, etc.), pumping and stirrer power input, cooling capacity, etc. [43].

With that issue in mind, the crude biowaste was in a first stage treated by winterization, a process usually applied in the industrial refinement of vegetable oils to remove waxes through crystallization [44]. The procedure promoted a considerable removal of the suspended waxes esters in the raw FDR (Figure 1A,B), along with a reduction in the viscosity (about 83%) and color of the epoxide (Figure 1C,D) under the same operating conditions.

FDR_w_ (FDR_winterized_) was then submitted to a typical epoxidation to increase thermal and oxidative stability, therefore improving the lubricant [45] and plasticizing properties [46,47]. To increase the hydroxyl content of the epoxidized oil, castor oil was incorporated (ratio 3:1), a non-edible natural polyol commonly used in the synthesis of bio-based plasticizers [48], thermosetting elastomers and cross-linked polymers [49,50,51]. 

The physicochemical properties of FDR_w_, derived epoxide, and of the formulated polyol, are shown in Table 1. All these products were characterized by acid and hydroxyl values as well as viscosity and density. The molecular weight and functionality were also determined for the polyol. 

FDR_w_ presented an average value of viscosity of 123 mPa·s, density of 0.962 g·cm^−3^, and hydroxyl (–OH) value around 45 mg KOH·g^−1^. The iodine value is parameter correlated to the percentage of unsaturation, i.e., the presence of reactive sites, the C=C bonds [52]. The obtained value of 130 g I_2_/100 g is typical of semi-drying vegetable oils such as soybean, corn, and sunflower oils [53]. 

The iodine value of the treated residue (FDR_w)_ showed the expected sharp decrease after epoxidation (from 130 to 32 g I_2_/100 g), i.e., a conversion of unsaturated bonds (*Iv_r_*) of about 76%. This variation is related to the breakdown of the double bonds in unsaturated fatty acids when oxidation reactions occur [54]. 

The epoxide presented a yield value of 89.2 % wt., based on FDR_w_ input and a relative conversion of the oxirane oxygen content (RCO) of about 61.4%.

The initial content in FDR_w_ hydroxyl groups (−OH = 42.5 mg KOH·g^−1^), derived from the presence of fatty alcohols and terpenes (e.g., farnesol), has increased after epoxidation (63.5 mg KOH·g^−1^). When castor oil was added (ratio 3:1), a value of 104.2 mg KOH·g^−1^ was achieved. The content of hydroxyl groups with the ability to form intramolecular H-bonding or the higher content of oligomers may contribute to the increase in the viscosity value observed on the formulated co-polyol [55], hence, the melting at relatively high temperatures under inert atmosphere. This operation allowed the reduction in water content (from 2.8 to 0.4%), reduction in viscosity, and the generally improved thermal properties (e.g., pour point and cloud point), seeking improved oxidation and hydrolytic stability [56].

After this treatment, the polyol became clear with a lower viscosity (15%) than the castor oil/epoxide mixture) with an enhanced flowability at room temperature. In fact, the values found for polyol’s viscosity and density (1350 mPa·s, 0.989 g·cm^−3^) were lower than the ones found for the epoxide (1600 mPa·s and 1.04 g·cm^−3^). 

The polyol presented an average molecular weight of 2978.9 Da and the obtained value of 5.5 hydroxyl groups per molecule is higher than the value found for castor oil of 2.7 hydroxyl groups per molecule [57], i.e., about the same order of magnitude as castor oil-based polyols obtained by thiol-ene reaction with 2-mercaptoethanol [58] and soybean oil polyols obtained using ethanol and ethylene glycol as hydroxylation agents [59]. 

The initial acid value for FDR_w_ (4.4 mg KOH·g^−1^), a reflex of the free fatty acids content (0.33 g/100 g lipids) [41,60], increased to 10.7 mg KOH·g^−1^ after epoxidation, probably due to the presence of residual carboxylic acid used in that step. After the addition of castor oil with a lower acid value (0.9 mg KOH·g^−1^), the value was reduced to 4.2 mg KOH·g^−1^.

### 2.2. Polyester Synthesis and Characterization

The polyester was synthesized by melt polycondensation between the FDR_w_-based polyol and azelaic acid, with citric acid (the crosslinker), in the presence of 1,4-butanediol and sorbitol as chain extenders and mechanical enhancers. The reaction took place under nitrogen purge and is catalyst and solvent-free to minimize any unwanted cytotoxicity, which is a prime parameter for biomedical applications [61].

The yield based on the initial input of polyol was about 78.6 % wt., a good result attesting to the economic viability of any industrial process.

Comparing both FDR and FDR_w_ structural profiles (Figure 2), no relevant changes are apparent. Visible throughout the entire process are the typical bands assigned to alkyl C-H bands at 2920, 2855, and 2973 cm^−1^. As expected, the bands at 831 and 863 cm^−1^ assigned to the highly reactive oxirane-ring structures (epoxy groups) already present in FDR_w_ are accentuated after epoxidation, evidence that some of the C=C double bonds remaining in the oily residue were broken and converted [62]. 

In the screening of the formulated polyol, there is an increment in the intensity of the bands at 3462 cm^−1^ and 1734 cm^−1^ assigned to –OH and C=O stretching, respectively, an expected result from the addition of the castor oil with higher hydroxyl content.

After esterification, the characteristic ester C=O stretching band at 1734 cm^−1^ suffers an exponential growth, and a new band emerged at 1171 cm^−1^ assigned to C-O stretching [63]. These are the signature bands of ester linkages typical of polyesters. The hydroxyl groups still attached to the carbon backbone of the cured polyester contribute to the hydrophilicity of the polymer [64]. 

The polyester presented a smooth and flexible surface, with an amber/brown color (Figure 3). 

To verify the surface wettability, the contact angle of the polyester was measured using water and squalane (Figure 4). The contact angle value stabilized within the time frame of 60 s for both fluids. For water (Figure 3), the contact angle showed a variation between 70.2° and 54.9°, thus confirming the hydrophilic character (moderate wettability) of this biopolymer [65]. Regarding the squalane oil, the change in the angle between 31.5 to 9.1° suggests good compatibility. That is a relevant fact since squalane and its natural counterpart, squalene, have been reported to be beneficial for skin health, exhibiting antioxidant, detoxifying, and regeneration activities. Squalane also acts as a drug carrier in both in vivo models and in vitro environments [66,67].

The droplet’s behavior indicated a spreading mechanism specifically observed with squalane, while absorption played a predominant role in the interface water/polymer surface, i.e., the surface wetting [68].

The polyester revealed a gel content (Table 2) in DMSO of 78.5% after 24 h, a sign of a relatively low cross-linking density that can be explained by the presence of unreacted oligomers [69]. 

The water absorption exhibited an average value of about 22.8% after 24 h. This value corroborates the hydrophilic nature of the polymer and the presence of hydroxyl terminations visible in the structural analysis [70].

Before applying any material in tissue engineering it is important to investigate their mechanical properties and thermal behavior. The results of the polyester tensile properties (Table 2) showed a tensile strength of about 0.19 MPa paired with an elongation at break with values between 102.1 and 153.0% (average 127.5%) and an average Young elastic modulus of 1.9 × 10^−3^ MPa. This behavior confirmed the polyester elastomeric properties [71]. 

Mechanical analysis of insulin-loaded PLGA nanofibers displayed elongation at a break of 164.3 ± 27.2% and tensile strength of approximately 2.87 ± 0.07 MPa, similar to human native skin [72]. A recent review [73] of commercial wound dressing with mechanical properties determined using tensile testing concluded that monolayered electrospun fiber dressings made of polyvinyl alcohol, carboxymethylcellulose, and alginate possessed a Young’s modulus between 0.24 and 0.95 MPa, with the total elongation varying between 68 and 134%.

Strategies to increase tensile strength can lie in increasing cross-linking density through curing conditions [74], or, for example, with the addition of glycol derivatives such as isosorbide [75] in the polymer composition. Additionally, the incorporation of natural fibers such as cellulose [76] or chitin [77] is also possible.

Analyzing the polyester thermogram (Figure 5), the material showed glass transition temperature (*T_g_*) taken at midpoint of −51.2 °C and a melting temperature (*T_m_*) of 169.6 °C, confirming a semi-crystalline nature, and non-glassy behavior at room temperature. An amorphous phase with a low glass transition temperature (*T_g_*) promotes the flexibility of the polymer network and physical net points that govern the form stability after elastic deformation [78].

### 2.3. Polyester in vitro Biocompatibility Study

Prior to the bio-application of any new material, it is necessary to check the cytocompatibility. The biocompatibility of this bio-based polyester was evaluated on human keratinocytes that were exposed to conditioned media with polymer discs (Figure 6). After an incubation period of 3 min and 24 h, the cell viability wasn’t visibly affected. 

The copolyester didn’t show any signs of toxicity when in contact with keratinocyte cells; hence, it was considered to be safe for skin applications. Keratinocytes are the predominant cell type found in the epidermis and the first to be affected by toxic substances through direct contact. Similar results were found by a polyester derived from palm fatty acids [79] and by a polyurethane film prepared with palm kernel oil-based polyester [80].

### 2.4. In Vitro Degradation Study

For successful tissue engineering applications, the polymeric biomaterial should degrade within a particular time span to allow the release of drugs in a sustained manner [81]. Hereupon, an in vitro degradation study was conducted under accelerated conditions (40 °C, RH = 75%). Two weeks after the beginning of the experiment (Figure 7), the polyester sample did not suffer mass loss; actually, there was an increment of about 41.4% with a visible expansion in the sample volume. Only after three weeks did the sample present a slight erosion around the edges, shown by a perceptible mass loss of 5.4%. In the following weeks, the mass loss was gradual; however, eight weeks later, the mass loss became exponential. After 11 weeks under accelerated conditions, i.e., one year in real-time, the bio-based polyester disintegrated completely in PBS solution. In the meantime, the pH value showed a slow reduction from the initial value of pH = 7.4 to pH = 6.7 until all semblance of integrity disappeared.

Since the diffusion of water into the polymer network preceded the hydrolytic degradation, the polyester degradation occurred in a two-step bulk erosion, a behavior similar to other aliphatic polyesters such as PLA and PCL [82,83,84]. In this type of erosion, after exceeding a critical value of water intake, the cleavage of the polymer chains, especially the hydrolytically unstable ester chemical bonds, results in water-soluble fragments; hence, the mass loss is fast, with a sudden release of degradation products [85]. Considering the results obtained so far, it is possible to conclude that lower cross-linking densities tend to promote a faster degradation [86]. 

There are several biopolymers extensively studied for applications in tissue engineering such as polycaprolactone (PCL), poly-l-lactic acid (PLA), and poly(glycolic acid) (PGA), whose biodegradation properties limit their clinical application [87,88]. The degradation of two years observed for PCL restricted its use for in vivo tissue engineering applications. High molecular weight crystalline PLA exhibited a degradation time of over 50 weeks [89,90].

The high degree of structural degradation observed with this particular bio-based polyester after one year, in closed containers and in vitro physiological conditions, suggests that this polyester will potentially undergo a higher hydrolytic degradation rate and biodegradation promoted by a microorganism-rich, moist, and warm environment such as a landfill [91,92].

### 2.5. Polyester Scaffold Properties

The design of a bio-based scaffold can be tuned to enhance mechanical properties and biodegradability through processing strategies directed to control pore size, pore quantity, and pore connectivity with the inherent change in density [93]. 

SEM micrographs (Figure 8) revealed the morphology of the scaffold porous surface along the structure of the pore walls. The surface of both scaffolds was found to be free from any irregularities such as cracking and delamination.

While 2D scaffolds exhibited a homogenous structure with well-formed pores (size 500 μm) only present on the surface, 3D scaffolds presented a longitudinal pore size gradient in the scaffold’s inner core. The difference in the distribution of the pores can be justified by the utilization of centrifugal forces in the production of the cylindrical scaffold [94].

### 2.6. In Vitro Dye Release Study

Several factors condition the drug delivery from a degradable polymeric matrix, such as the degree of the polymer hydrophilicity, the surface erosion and cleavage of polymer bonds within the matrix, and the diffusion mechanism of the entrapped drug [95].

The capacity of the prepared polyester to adsorb and release a drug and the rate of that delivery was studied using two dyes. 

Curcumin (CRC) was selected for its recognized potential as a wound healing agent along with anti-infectious, anti-inflammatory, and antioxidant properties [96,97]. Quite recently, curcumin-loaded delivery systems for wound healing applications were the subject of an updated review [98].

Additionally, to evaluate the capacity of the manufactured polyester to entrap water-soluble molecules, Rhodamine B base (RBB) was selected as a model [51,99]. This dye was used with the same purpose in the case of polylactic acid microchamber arrays and liquid resins in 3D-printed medical devices [100,101].

In the end, two data sets were obtained using the manufactured scaffolds. Figure 9 illustrates the release curves for both dyes and the plots regarding the application of the Korsmeyer–Peppas model. In Table 3 are the compiled results. 

Concerning RBB dye, the 3D scaffolds presented a loading capacity after 7 h of 1.60 mg dye·g^−1^, and within 48 h about 29.3% of the dye was released. Nevertheless, when the contact water was replaced after 10 days, the dye was still being released to the medium. Regarding CRC, the 2D structures revealed a loading capacity of 0.64 mg dye·g^−1^ after 7 h, and in the same period about 50.4% of the dye was released in the ethanolic contact solution. No further release occurred when replacing the contact solution, although the polymer still presented some coloration.

After application of the Korsmeyer–Peppas model Equation (11) to the release data set, correlation coefficients (R^2^) were 0.987 (RBB) and 0.996 (CRC), respectively, indicating a good adjustment between the model and the experimental data. From the data fitting, the transport constant for CRC was *k* = 29.1 h^−1^ and the transport exponent *n* = 0.31 and for RBB, *k* = 11.9 h^−1^ and *n* = 0.24 (Table 3). Since its *n* < 0.5, the diffusional model is a quasi-Fickian model, i.e., a non-swellable matrix-diffusion [102].

These results concur with the curcumin release from poly(lactic acid)/polycaprolactone electrospun fibers which also followed a diffusion-controlled mechanism, with a good fit to the Korsmeyer–Peppas model, and a diffusion coefficient, *n* ≤ 0.5 [89]. A poly(lactic acid)/polycaprolactone polyester, loaded with a complex of curcumin and β-cyclodextrin with an efficacy release of 20% was developed with potential use for wound dressings, drug delivery, and regenerative systems [90]. The Korsmeyer–Peppas model has been widely used to describe bioactive compound release from polymers, namely grapeseed extract from nanofibrous membrane made with polylactic acid and polyethylene oxide [92], and rutin release from cellulose acetate/poly(ethylene oxide) [93]. 

The manufactured 3D scaffolds showed a lower release rate and a higher loading capacity to entrap hydrophilic molecules than the 2D structures for the lipophilic molecule. This information suggests that this polyester could potentially be used to deliver drugs with hydrophilic characteristics into the skin, as in the case of hyaluronic acid and ascorbic acid in PCL [103].

## 3. Materials and Methods

### 3.1. Reagents

*β*-Farnesene distillation oily residue (FDR) was kindly provided by Amyris and sugarcane-based Squalane (Neossance^TM^) from Aprinnova LLC, Emeryville, CA, USA. Formic acid 99% was supplied by VWR; sodium chloride, ethanol 96% (EtOH), ethyl acetate, hydrogen peroxide (35%), sodium hydrogen carbonate, and citric acid monohydrated (CA) from Labchem. Pressurized nitrogen of high purity was supplied by Gasin. Azelaic acid (AzA), sorbitol (S), and 1,4-Butanediol (BTN) were all supplied by Sigma. Acofarma supplied castor oil containing 85–90% of OH-bearing ricinoleic acid (C18:1). This non-edible oil was characterized by an acid value of 0.9 mg KOH·g^−1^, hydroxyl value (166 mg KOH·g^−1^), iodine value of 83 gI_2_/100 g, 0.11% of water, and viscosity of 986 cp (20 °C).

### 3.2. Epoxide Synthesis and Polyol Formulation

FDR pre-treatment (FDR_w_) Crude FDR was treated by winterization, a procedure where waxes are removed through crystallization with a suitable solvent at low temperatures. FDR was thoroughly mixed with ethanol 96% (mass ratio EtOH: FDR = 4:1) for 30 min, followed by overnight decantation at 4.0 ± 2.0 °C in a refrigerator (Thermo Scientific^TM^, Waltham, MA, USA) and additional clarification by centrifugation (Thermo Scientific^TM^, Heraeus Megafuge 16 R). The ethanol was recovered to be recycled in a rotary evaporator (Heidolph, Hei-Vap Precision, Schwabach, Germany). The clarified product (FDR_w_) was reserved for further analysis and used in the next stage. 

*Epoxidation (Epoxide)* was carried out with in situ performic acid according to [104] with slight modifications. FDR_w_ (200 g), ethyl acetate (100 g), and formic acid (12 g) were weighed in a closed borosilicate vessel with a three-way lid and placed on a heating/stirring plate (Hei-Tech, Heidolph Instruments GmbH & Co.KG, Schwabach, Germany) with temperature control. After a few minutes of constant stirring, refrigerated hydrogen peroxide (320 g) was slowly poured to avoid exothermic foaming. The epoxidation took place at 85 °C for 3 h with constant stirring, followed by decantation of the organic phase in a separatory funnel and a neutralization/washing operation with water and sodium bicarbonate solution (10%). The epoxide was then dried overnight at 80 °C in a vacuum oven (Binder, model VD23).

Polyol formulation (Polyol) FDR_w_-based epoxide (300 g) was mixed with castor oil (100 g) and heated at 160 °C for 7 h under nitrogen inert atmosphere with constant stirring.

### 3.3. Polyester Synthesis 

The selected reagents were the formulated polyol (P), azelaic acid (AzA, saturated diacid); citric acid (CA, cross-linker); and 1,4-butanediol (BTN) and sorbitol (S) both acting as chain-extender/mechanical enhancers. The prepolymer was prepared with a reagent mass ratio P: Aza: CA: BTN = 6:4:3:1:1, using an eco-friendly solvent and catalyst-free melt-polycondensation method [51], under nitrogen purge. In the first phase, all reagents except sorbitol were melted together at 160 °C for 1.5 h with constant stirring. After that period, the sorbitol was added, and the reaction was prolonged for 30 min. The reactor content was then quickly poured into a PTFE mold, and the curing was carried out in a vacuum drying oven for 3 days at 140 °C. The result was a thermoset copolyester.

### 3.4. Scaffolds Production

The scaffold production was carried out by salt leaching, an easily available and inexpensive technique used here as a proof-of-concept. A known quantity of sieved (Sieve shaker AS200, Retsch GmbH, Düsseldorf, Germany) sodium chloride crystals, with particle size between 315 and 500 μm, was mixed with the prepolymer right before casting. The content was partially poured into the square PTFE mold, resulting in a flat 2D membrane with a thin, porous layer. To obtain cylindrical structures with a different pore distribution (noted as 3D), another fraction was poured into polypropylene tubes (10 mL) and quickly centrifuged at 10,000 rpm. Both molds were placed in the vacuum drying oven and cured at 140 °C for 3 days. After cooling to room temperature, the tubes were cut into slices of similar dimensions and cooled down in the refrigerator at 4.0 ± 2.0 °C to allow the polymer detachment. The retained salt porogens were then dissolved with lukewarm deionized water, and to control the leaching process, conductivity measurements were periodically taken (pH/mV Mettler-Toledo Seven Excellence Multiparameter), until the registered value was similar to the solvent conductivity.

### 3.5. FDR, Epoxide, and Polyol Characterization

Density was determined by the pycnometer method following ASTM D792-08.

Dynamic viscosity was determined using a vibrational viscometer (SC-10, Scansci, Vila Nova de Gaia, Portugal). This parameter was evaluated in the initial and intermediate products (FDR_w_, epoxide, and polyol). 

Epoxidation and hydroxylation yields were based on the FDR_w_ initial input and were determined by Equation (1).
(1)$$Yield\;\left(\%\right)=\frac{m_{f}}{m_{i}}\times 100$$
where *m_f_* is the weight of the final product, and *m_i_* the weight of the initial input_._

Iodine value variation or, just another way of saying, the conversion of unsaturated bonds (*I_vr_*) was determined by the Equation (2):(2)$$Ivr\;\left(\%\right)=\frac{\left(I_{v0}-I_{vf}\right)}{I_{v0}}\times 100\;$$
where *I_v_*_o_ is the initial (FDR_w_) iodine value, and *I_vf_* is the iodine value after epoxidation stage.

Oxirane oxygen content (*O.O.C.*), was determined by titrimetric analysis following the HCl-acetone ultrasonic assisted method [105] and computed by Equation (3) after [106,107].
(3)$$O.O.C.\;exp.\;\left(\%\right)=\frac{\left(V_{0}-V\right)\times M\times 16}{\left(1000\times W\right)}\times 100=\frac{\left(V_{0}-V\right)\times M\times 1.6}{W}\;$$
where *V*_0_ = Volume of NaOH solution required for the blank; *V* = Volume of NaOH solution required for the sample; *M* = NaOH solution molarity (mol/L); *W* = sample weight (g).

The relative conversion of the oxirane oxygen content (RCO) was determined by the expression (4):(4)$$\mathrm{RCO}\;\left(\%\right)=\frac{O.O.C.\;exp.}{O.O.C\;theor.}\times 100$$
where
(5)O.O.C. theoretical (%)=[(IV02AI2)(100+IV02AI2)×AOx]×AOx×100
where, *A_I_*_2_ = iodine atomic mass (126.9); *A_Ox_* = oxygen atomic mass (16.0); *I_vo_* = iodine value for the initial sample (130 gI_2_/100 g).

Acid and hydroxyl (–OH) values, were determined by titrimetric analysis following ASTM D4662-08 and ASTM D1957-86, respectively.

Structural analysis was performed using Fourier Transform Infrared Spectroscopy (FTIR) to envision the chemical structure of all compounds in the near infrared range, between 4000 and 700 cm^−1^. The equipment used was a Perkin Elmer FT-IR spectrophotometer, fitted with Pike Miracle ATR accessory containing ZnSe crystal.

Polyol molecular weight distribution was determined by size-exclusion chromatography using a high performance liquid chromatograph (model 1260 Infinity II, Agilent Technologies, Santa Clara, CA, USA) attached to an Evaporative Light Scattering Detector (ELSD, 1290 Infinity II, Agilent Technologies, Santa Clara, CA, USA) with evaporation temperature at 70 °C and nebulization at 65 °C, using nitrogen as nebulizing gas coupled to a TSK gel GMHxL column for insoluble polymers. The isocratic analysis was carried out with tetrahydrofuran as the mobile phase; flow rate of 0.6 mL·min^−1^; sample concentrations of 20–25 mg·mL^−1^ dissolved in THF and injection volumes of 20 μL. The molecular weight was estimated by a calibration curve of polystyrene-standards 400–303,000 Da (Agilent (Waldbronn, Germany). 

Polyol functionality (*f*), defined by Equation (6), was calculated based on the average molecular weight (*M_w_*) and equivalent weight (E) was determined using –OH number.
(6)$$f=\frac{M_{w}}{E}\;\mathrm{where}\;E=\frac{\left(1000\times 56.1\right)}{\left[-\mathrm{OH}\right]}$$

### 3.6. Polyester Characterization 

Static contact angle of the liquid−solid interface was determined using water, squalane oil, and cured polyester. This property was measured by the sessile drop method at room temperature using the tensiometer (Attention^®^ Theta Lite, Biolin Scientific, Gothenburg, Sweden) with OneAttension software version 4.0.2. Before the experiment, the polyester film was cleaned, dried, and cut into 20 × 20 mm squares. A small drop of fluid (20 μL) was placed in the polymer surface and the angle made with the tangent was recorded. Since for biopolymers, no contact angle equilibrium is expected, but rather a pronounced variation during the first 60 s (Farris et al., 2011), the analysis was focused on that time window. The measurements were carried out at three different positions on the sample surface and at three different moments in time.

Gel content was calculated following [75], as the fraction of the polyester insoluble in dimethyl sulfoxide (DMSO). Cured polyester strips were cut out with dimensions 40 × 10 × 2 mm, weighed (*W_i_*), and immersed in the solvent for 24 h at room temperature. The insoluble fraction was filtered in a pre-weighed filter paper and dried under vacuum at 80 °C overnight. *W_e_* is the recorded weight of the dried samples after being extracted. The gel content determined using Equation (7) was taken as the average of three samples.
(7)$$Gel\;content\;\left(\%\right)=\frac{W_{e}}{W_{i}}\times 100$$

Water absorption was determined following ASTM-D 570-98, replacing water by PBS solution to mimic physiological conditions. Dried disk samples with a known weight (*W_d_*) were immersed in PBS (pH = 7.4 ± 0.2) solution at 37 °C for 24 h. After that, the samples were withdrawn from the liquid; the excess of solution was removed from the surface with filter paper and weighed (*W_t_*). The water absorption was determined by Equation (8) and is expressed as a percentage (%).
(8)$$Water\;absorption\;\left(\%\right)=\frac{W_{t}-W_{d}}{W_{d}}\times 100$$

Mechanical performance was evaluated using a Texture Analyser (model TA.XT plus C) with data acquisition and treatment software Express Connect v7.3 (Texture Technologies Corp. and Stable Microsystems Ltd., Hamilton, MA, USA). Several longitudinal strips with dimensions 100 × 10 × 2 mm were cut from the polyester mat and each one was attached to miniature tensile grips. The experiment was carried out at room temperature using a 30 Kg load cell with a strain rate of 2 mm·min^−1^. The elastic modulus (Young modulus) and % elongation were the determined properties. 

Thermal analysis was performed by differential scanning calorimetry (DSC) using a calorimeter (DSC 204, NETZSCH GmbH Co.Holding KG, Bayern, Germany), and nitrogen as the purge gas (40 mL·min^−1^). Approximately 2 mg of each sample was placed in aluminum pans and the thermal properties were recorded between −70 and 300 °C at 10 °C ·min^−1^ to observe the melt (*T_m_*) and glass transition (*T_g_*) temperatures. The latter was measured on the second heating ramp to erase the thermal history of the polymer.

In vitro cytotoxicity of the cured polyester was evaluated by indirect contact. The human keratinocyte cell line HaCaT (CLS) was kept in culture in DMEM media (Gibco, Waltham, MA, USA) supplemented with 10 % FBS (Gibco) and 1% antibiotic (Gibco) at 37 °C, with 5% CO_2_ in a humidified atmosphere. To perform assays with cell lines, polymer discs were sterilized by immersion in ethanol for 1 h and briefly washed with sterile PBS prior to the experiment. Polymer discs were incubated with media in 24 well plates for 3 min and 24 h at 37 °C, with 5% CO_2_ in a humidified atmosphere. Then, a previously seeded 96-well plate with HaCaT at 1 × 10^4^ cells/well was incubated with the conditioned media in quadruplicate. Wells with only conditioned media (without cells) were used to subtract a possible influence of the samples in the PrestoBlue fluorescence signal. Cells treated with 10% DMSO (dimethyl sulfoxide) were used as a negative control. After 24 h of exposure to the conditioned media, cytotoxicity was evaluated by the metabolic inhibition using a PrestoBlue assay (Thermo Fischer), according to the manufacturer’s instructions. PrestoBlue reagent was added to the media and incubated for 2 h at 37 °C, with 5% CO_2_ in a humidified atmosphere. The fluorescence signal was read in a Synergy H1 microplate reader (BioTek, Winooski, VT, USA). The results are expressed in percentage of cell viability as compared with a control (cells with plain media). At least two independent experiments were performed.

In vitro degradation in physiological conditions was determined in accordance with ISO 175:2010, in a climatized chamber (Climacell 11 Ecoline, MMM Medcenter Einrichtungen, GmbH, München, Germany) under accelerated conditions, i.e., with a temperature of 40.0 ± 2.0 °C and relative humidity (RH) of 75 ± 5%. Pre-weighed (*m_i_*) dried disks of very similar dimensions (diameter = 10.7 ± 0.01 mm and thickness of 1.18 ± 0.02 mm) were individually placed in closed glass vessels containing 10 mL of PBS solution (pH of 7.4 ± 0.1) and placed in the controlled environment chamber. Weekly, one of the vessels was retrieved and the disk was dried at 50 °C for 48 h. The sample final weight (*m_f_*) was registered, along with the pH of the remaining saline solution. The mass loss (%) related with sampling time was determined using Equation (9).
(9)$$mass\;loss\;\left(\%\right)=\frac{m_{i}-m_{f}}{m_{i}}\times 100$$

### 3.7. Scaffold Properties

Surface morphology was visualized using a JEOL-5600 LV Scanning Electron Microscope (Tokyo, Japan) from JEOL, Japan, equipped with a SPRITE HR Four Axis Stage controller (Deben Research). Samples were placed directly on double-sided adhesive carbon tape (NEM tape, Nisshin, Japan), placed on metallic stubs covered with adhesive carbon tape (NEM tape, from Nisshin, Japan) and coated with gold/palladium using a Sputter Coater (Polaron, from Bad Schwalbach, Germany). All observations were performed in high-vacuum with an acceleration voltage of 30 kV, at working distance of 12–13 mm and a spot-size of 20. 

In vitro dye adsorption/release was monitored with a UV-Vis spectrophotometer (Shimadzu model UV-1900), using water-soluble dye (Rhodamine B base, RBB) on 3D scaffolds and hydrophobic curcumin (CRC) in 2D scaffolds to study the behavior of previously prepared scaffold disks. With that purpose, 100 mL of each dye were prepared dissolving a certain quantity of RBB in deionized water, and CRC in PBS ethanolic solution (EtOH 35 % *v*/*v*). Sets of pre-weighed and dried scaffold disks were immersed in 20 mL of dye solution at 20.0 mg /mL and 7.2 mg/mL for RBB and CRC, respectively, and the vessels were placed in an incubator shaker (Innova 40, series S) at 30 °C and 120 rpm for 7 h. 

After that period, the scaffolds were removed, the excess liquid absorbed with paper filter and the disks dried overnight at 50 °C. The 3D scaffolds (RBB dye) were then placed in 20 mL of aqueous solution and the 2D disks (CRC loaded) were immersed in 20 mL of the PBS ethanolic solution. The amount of dye released was quantified hourly until equilibria conditions were achieved or within an established period of 48 h.

The concentration of the entrapped dyes was determined using standard calibration curves in the linear range at the wavelength of maximum absorbance, (RBB, 544 nm) and (CRC, 430 nm).

The loading capacity (L.C.) was calculated from the following Equation (10):(10)$$Loading\;capacity\;\left(\frac{mg\;dye}{g\;scf}\right)=\frac{m_{di}-m_{df}}{m_{scf}}\times 100$$

*m_di_* and *m_df_* are the initial and final amount of dye in the contact solution (mg), and *m_scf_* the scaffold initial weight.

To evaluate the diffusional mechanism regarding the release kinetics of both dyes, we applied the Korsmeyer–Peppas model [108]. This empirical Equation (11) allows us to analyze both Fickian and non-Fickian release of drug from swelling and non-swelling polymeric delivery systems [102].
(11)$$Ln\left(\frac{M_{t}}{M_{\infty }}\right)=Ln\left(k\right)+n\;Ln\left(t\right)$$

In this equation, M_t_/M_*∞*_ is the fraction of dye (expressed in % of release) delivered at time *t*; *k* is the transport constant (dimension of time^−1^); and *n* is the transport exponent (dimensionless). The release constant *k* provides mostly information on the drug formulation such as structural characteristics, whereas *n* is important since it is related to the release mechanism (i.e., Fickian diffusion or non-Fickian diffusion) [109]. 

## 4. Conclusions

In this study, a fermentation by-product derived from the production of β-farnesene was used as feedstock in the synthesis of an aliphatic copolyester by polycondensation.

This polymer exhibited interesting elastomeric properties described by the relevant elasticity and low Young modulus along with low *T_g_*, and high *T_m_*. 

The hydrophilic nature of this polymer was translated to relevant water uptake, followed by hydrolytic degradation by bulk erosion in saline solution within one year. This behavior suggests the possibility of easier degradation in landfill conditions (acidic, warm, and microorganism-rich), an added benefit concerning environmental impact. 

The bio-based polyester showed evidence of non-toxicity to human keratinocyte cells, and dye release studies of the polyester scaffold showed a diffusion-controlled mechanism compatible with drug delivery applications.

These properties show a promising possibility, for instance, in the production of a nanofiber layer designed to carry bioactive molecules to integrate into a dressing for wounded or burnt skin tissue. Further studies will be necessary.

Considering these conclusions, this polymer offers a sustainable and eco-friendly alternative for the potential use of controlled release of active principles for wound dressing applications.

## Figures and Tables

**Figure 1 ijms-24-04419-f001:**
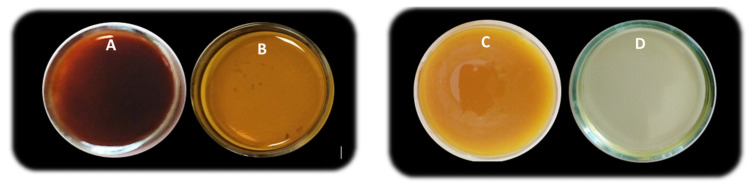
Picture of FDR (**A**), FDR_w_ (**B**), epoxide obtained from FDR (**C**), and epoxide obtained from FDR_w_ (**D**).

**Figure 2 ijms-24-04419-f002:**
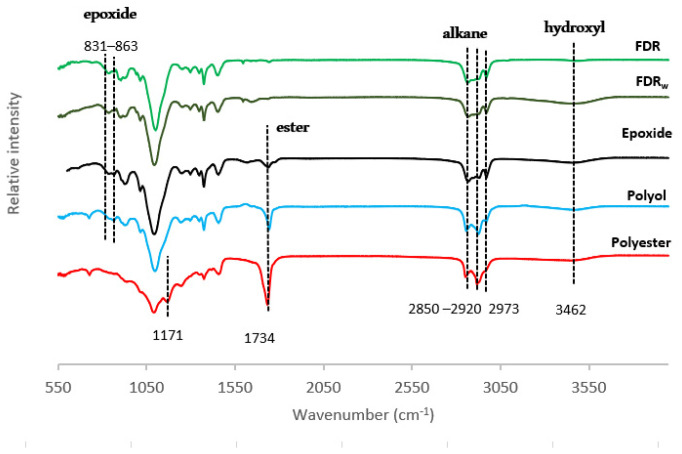
FTIR spectra. Structural evolution from FDR_w_ to polyester.

**Figure 3 ijms-24-04419-f003:**
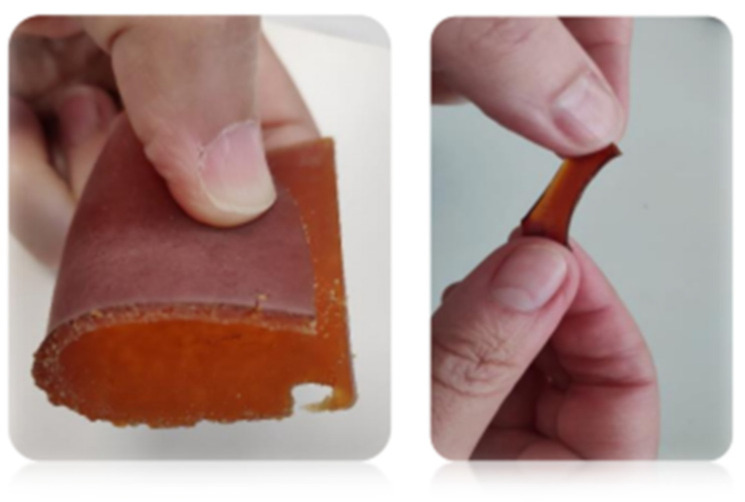
Picture of the flexible polyester obtained using the FDR.

**Figure 4 ijms-24-04419-f004:**
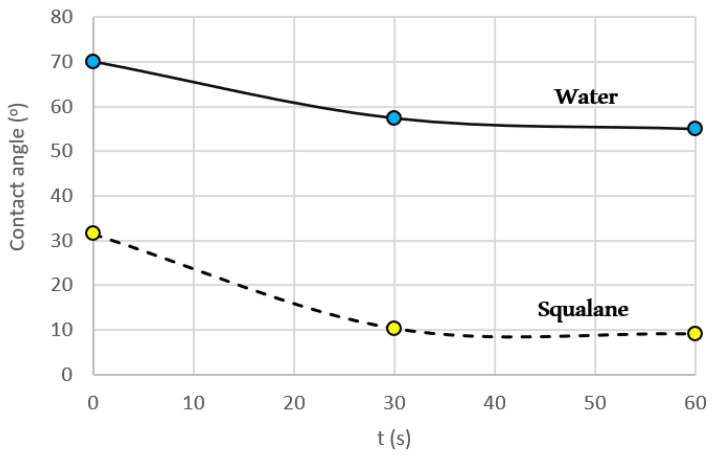
Static contact angle variation, experimental results within 60 s, for water (•) and squalane (•).

**Figure 5 ijms-24-04419-f005:**
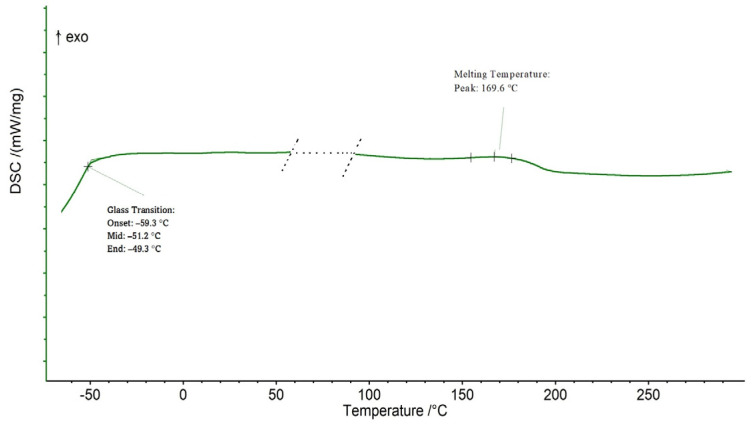
Polyester thermal performance (DSC).

**Figure 6 ijms-24-04419-f006:**
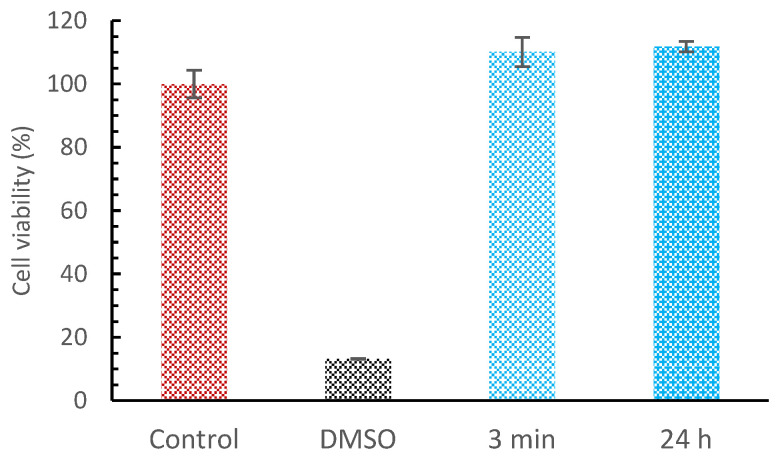
Polyester in vitro cytotoxicity or cell viability of HaCaT cells exposed to media conditioned with polymer discs, for 3 min and 24 h.

**Figure 7 ijms-24-04419-f007:**
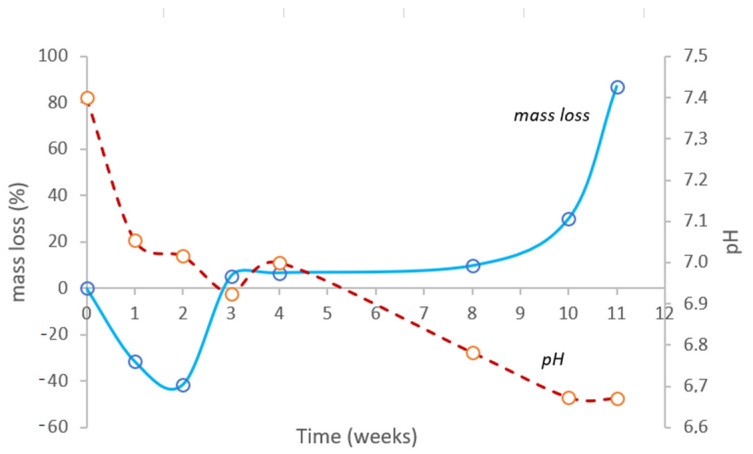
In vitro degradation under accelerated physiological conditions (PBS solution, 40 °C, RH = 75%) represented by mass loss. pH variation of solution is represented by the dash line.

**Figure 8 ijms-24-04419-f008:**
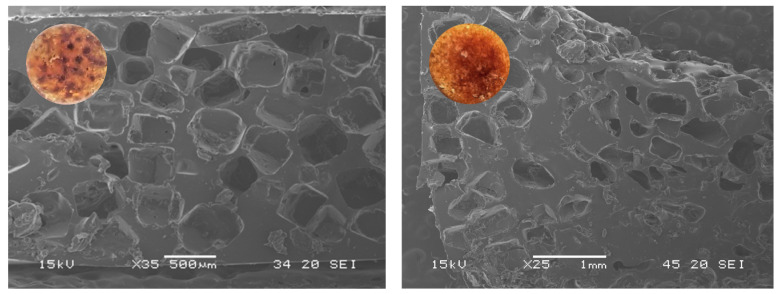
Scaffolds Micrographs (SEM analysis). On the left, the 2D scaffold sample and on the right, the 3D scaffold sample.

**Figure 9 ijms-24-04419-f009:**
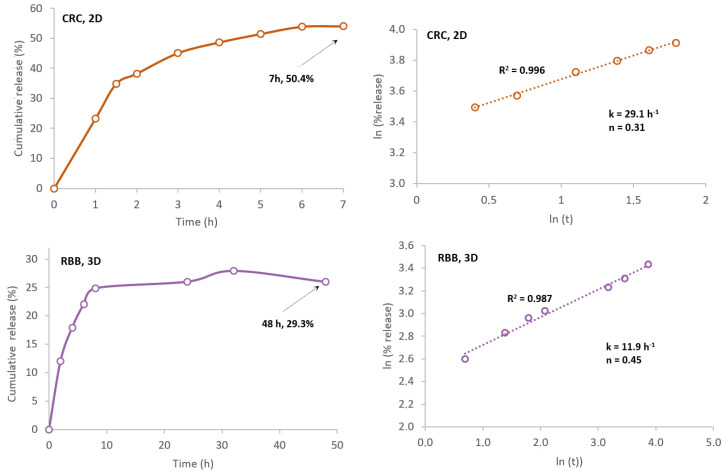
In vitro release profiles of RBB and CRC dyes and the plotted results obtained by application of Korsmeyer–Peppas model.

**Table 1 ijms-24-04419-t001:** Materials’ chemical-physical characterization (FDR_w_, Epoxide, and Polyol).

Properties	FDR_w_	Epoxide	Polyol
Iodine value (g I_2_/100 g)	130.0 ± 7.5	32.0 ± 0.5	32.0 ± 0.5
Acid value (mg KOH·g^−1^)	4.4 ± 0.9	10.7 ± 0.2	4.2 ± 0.3
–OH Value (mg KOH·g^−1^)	42.5 ± 2.0	63.5 ± 2.0	104.2 ± 12.5
Viscosity (mPa·s, T = 23 °C)	123 ± 10	1600 ± 12	1350 ± 34
Density (g·cm^−3^, T = 23 °C)	1.0	1.0	1.0
*M_w_* (Da)	-	-	2978.9 ± 17.6
Functionality (–OH/molecule)	-	-	5.5
Reaction Yield (wt%)	-	89.2	-

**Table 2 ijms-24-04419-t002:** Polyester properties: gel content, water absorption, and tensile properties.

Property	Unit	Value
Gel content (DMSO)	(%)	78.5 ± 1.7
Water absorption	(%)	22.8 ± 4.0
Young’s Modulus *	(MPa)	1.9 ×10^−3^ ± 8.3 × 10^−5^
Tensile Strength *	(MPa)	0.19 ± 0.01
Elongation at break *	(%)	127.5 ± 25.5

* Results are expressed as average ± standard deviation (*n* = 6).

**Table 3 ijms-24-04419-t003:** In vitro dye adsorption/release, sorption, and kinetic parameters.

Scaffold	Dye Type (Solvent)	L.C. (mg_dye_·g^−1^)	Release (%)	k (h^−1^)	*n*
3D	RBB _(aqueous)_	1.6	29.3 (48 h)	11.9	0.45
2D	CRC _(PBS/EtOH)_	0.64	50.4 (7 h)	29.1	0.31

## Data Availability

This study did not report any data.

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
