# Peer review of "Synthesis of Bio-Based Polyester from Microbial Lipidic Residue Intended for Biomedical Application"

_ijms, 2023, doi:10.3390/ijms24054419_

Round 1

Reviewer 1 Report

­­In this manuscript, the authors synthesize a bio-based polyester polymer for drug delivery applications. A variety of mechanical, chemical, and in vitro experiments were performed to assess its suitability for drug delivery. The manuscript is well written and the approach and conclusions are generally well justified, except for an adequate discussion of how this polyester scaffold design could be of use in wound healing applications. The industrial synthesis methods seem scalable which adds to its appeal. This manuscript seems suitable for publication in IJMS.

Questions and Comments:

(1) There is no x-axis in Figure 4.

(2) In Figure 6, where are the individual data points? How are these smooth curves produced?

(3) The applicability of the porous scaffold studied by the authors should be discussed in the context of wound healing. How would this device be used for wound healing?

(4) Is it possible to present the drug release curves underlying the data presented in Table 3? This would add to the understanding of the reader.

(5) It would have been much more interesting to have seen this polymer used to produce a nanofiber scaffold, which could actually facilitate wound healing.

Author Response

First, at all, author’s would like to thank all the suggestions given by the reviewers under supervision of the editor, author’s have carefully undertaken the revision requests for your reconsideration. All the changes made in the manuscript are highlighted at blue color directly in text and the identified in the reviewer’s letter with the line number.

Reviewer 1

Comments and Suggestions for Authors

­­Comment 1: In this manuscript, the authors synthesize a bio-based polyester polymer for drug delivery applications. A variety of mechanical, chemical, and in vitro experiments were performed to assess its suitability for drug delivery. The manuscript is well written, and the approach and conclusions are generally well justified, except for an adequate discussion of how this polyester scaffold design could be of use in wound healing applications. The industrial synthesis methods seem scalable which adds to its appeal. This manuscript seems suitable for publication in IJMS.

Questions and Comments:

Comment 2: There is no x-axis in Figure 4.

Answer: As suggested by the reviewer the x-axis was added now to Figure 5.

Comment 3: In Figure 6, where are the individual data points? How are these smooth curves produced?

Answer: The data points were included in the revised article and can be seen now on Figure 7.

Comment 4: The applicability of the porous scaffold studied by the authors should be discussed in the context of wound healing. How would this device be used for wound healing?

Answer: That subject was addressed in the conclusions of the revised article (lines 584-586).

Comment 5: Is it possible to present the drug release curves underlying the data presented in Table 3? This would add to the understanding of the reader.

Answer: The release curves for both dyes was presented in the revised article, in Figure 9.

Comment 6: It would have been much more interesting to have seen this polymer used to produce a nanofiber scaffold, which could actually facilitate wound healing.

Answer: The presented technique was used as a proof-of-concept and besides that the salt leaching technique is a cheap and easy has procedure. The production of nanofiber scaffold as suggested by the reviewer is an interesting approach that will facilitate the application for wound healing, however this technique wasn’t available in our facilities and the process itself still requires optimization.

Reviewer 2 Report

This paper presents a study of the synthesis and properties of polyester based on biowaste, a microbial oil, and castor oil to produce a porous scaffold for wound dressing. The manuscript contains some interesting results but it needs a major revision especially in the synthesis section.

First, I don't understand what is the composition of the FDR used in the synthesis. Authors claimed that it contains "hydrocarbons (43.29 g/100 g lipids), a small percentage of complex lipids such as tri- and di-glycerides along with farnesol and terpenes". These data are taken from Ref. 46. But authors also claimed that the winterization was applied to purify and provide some uniformization of this reagent. And what was the composition of FDR after winterization? What were the components for epoxidation: unsaturated hydrocarbons, farnesene, farnesol, glyderides?

Second, the question concerning the synthesis of polyol and the description of its IR-spectrum.  

L.149-151. Authors wrote: "This is a result of the functionalization of the epoxide by the castor oil fatty acids, mainly ricinoleic acid which has specifically two functional groups, ester, and hydroxyl". – ricinoleic acid does not contain ester groups. It is an acid. If authors mean that castor oil (which contains up to 90% ricinoleic acid triglycerides) is a functionalizing agent, they should rewrite this sentence. And another question – how the authors imagine the functionalization of epoxide with castor oil. What is the reactant interacting with epoxy groups – triglycerides of ricinoleic acid (which contain hydroxyl groups in their structure) or free fatty acids also present in castor oil? But I did not observe the decrease in the intensity of the band corresponding to epoxide ring (831-863cm-1). And taking into account that the polyol synthesis was carried out simply by mixing epoxide with castor oil at elevated temperature without any separation or purification I do not see any strong evidence of the polyol synthesis.

Third. Authors wrote in the Conclusion that "…microbial oil … was used as feedstock in the synthesis of an unsaturated aliphatic and linear copolyester". How did authors synthesize linear copolyester via using polyol, azelaic diacid and cross-linker? The use of polyol (by the way, what is the average number of OH-groups per molecule of polyol?) should give branched product and cross-linker use results in the formation of polymer network.

So, the main problem in the synthesis section is the lack of understanding of which substances are the initial reagents and which are formed as a result of the synthesis. Therefore, the authors must adjust the synthesis methods and perform additional analyzes to be sure of the composition and structure of the initial and final product. Besides, general schemes of occurred reactions should be added to the text.

Other remarks:

L. 104. "…the iodine value, presented the expected sharp decrease after FDR epoxidation from 130 to 32 mg KOH.g-1…" Iodine value is expressed as the mass of iodine per 100g of substance, not mgKOH.

L.116-118. – " The influence of reaction conditions was clearly demonstrated by the synthesis of different polyols based in soybean oil through oxirane cleavage with water catalyzed by perchloric acid, with -OH values of 160, 240, and 285 mg KOH.g-1 [52]." – I don't understand the sense of this sentence. If authors wanted to demonstrate the influence of the reaction conditions onto hydroxyl value, it should be pointed in the text what reaction conditions caused such change in -OH value. 

L.122 and Table 1. PDI value of polyol presented in this table and in the text is equal to 0.99. How can it be? PDI=(Mw/Mn). And weight average molecular weight is always larger than number-average. Should I believe to other data after that? Another question: what molecular weight is presented in the table: Mw or Mn? And I have never seen molecular weight determined to tenths of a Dalton via liquid chromatography.  

L.131-132. "The reduction to the final value (4.2 mg KOH.g-1) for the polyol, is an improvement of the hydrolysis resistance" – I don't understand what authors would like to say in this sentence. Please rephrase it. In my opinion, the decrease in acid number can be explained by the introduction in the system of castor oil with rather low acid value (0,91 mg KOH/g - doi:10.1021/acsomega.0c05810)

L. 147-148. "In the polyol screening, the peaks exhibited at 2930 cm−1 and 2855 cm−1 assigned to alkane groups are better defined and are visible in the increasing intensity of the bands at 3462 cm-1 and 1734 cm-1, assigned to -OH and C= O stretching, respectively". I don't understand "alkane groups are better defined" – better than where or better than what? And how peaks assigned to alkane groups " are visible in the increasing intensity of the bands at 3462 cm-1 and 1734 cm-1"? Is it simply poor English or something else?

L.189. "…a Young elastic modulus between 1.9×10-3 and 2.2×10-3 MPa." – The data of Table 2 witness that Young elastic modulus is between 1.6×10-3 and 2.2×10-3 MPa

L. 325. "Castor oil (non-edible oil) with 85-90 % content in ricinoleic acid" – Castor oil does not contain 85-90% ricinoleic acid. It contains 85-90% triglycerides of ricinoleic acid. Authors should determine acid value and maybe hydroxyl value of castor oil to be sure of the free acid content in the oil composition. Because the acid number for castor oil ranges usually from 0.14 to 1.97 mgKOH/g oil (doi:10.1590/fst.19620), while acid number for ricinoleic acid is 175-190 mg KOH/g

L. 353-354. "Epoxidation and hydroxylation reaction yields were based on the FDR initial input were determined according to equation (1) and expressed in percentage: Y=(mf/mi)x100, where, mf is the epoxide or polyol mass and mi is …FDR initial mass". – I don't understand how the polyol mass was determined if after the hydroxylation reaction "…no further purification was necessary." (L. 352). In such case the yield will be higher than 100%. How did authors separate final product from castor oil?

L. 421. Gel-content determination. Did the authors grind or cut the samples to determine the gel-fraction? What temperature was used for extraction?

L. 438. Equation for water absorption calculations. %=(Wt-Wd)/Wt. Water absorption is calculated as the difference between the weight of soaked sample and dry initial sample divided by weight of initial weight of dry sample. So, the equation 5 should be as follow: %=(Wt-Wd)/Wd.  

Some references are out of order. For example:

P.5, L. 167. (Menzies and Jones, 2010). – I didn't find this reference in the References section.

P.9, L. 341. "Epoxidation (Epoxide) was performed according to [100] with slight modifications" – Reference [100] does not contain any technique for epoxidation. Please double check this reference.

P.13., L. 514. (Korsmeyer et al., 1983) – I didn't find this reference in the References section.

English also should be improved.

There are many mistakes (such as word order, comma use, wrong words etc.) that make it difficult to understand the tex. See some of them below please.

L. 73-74. "Several natural synthetic polymers have been used for wound healing applications since they join properties, such as biocompatibility and/or biodegradability…" – "…they combine properties…" sounds better.

L. 76-77. "…since to achieve the required properties for tissue regeneration, biopolymers, have high production costs." – Please rephrase this sentence.

L. 117. "…based in soybean oil…" – in English should be "…based on soybean oil…".

L.154. "The contact angle value stabilized within the time frame of 60 s, for both fluids, and for water, showed (Figure 3) a variation between 70.2° and 54.9°" – Please double check comma use.

L. 167-171. "Regarding the squalane oil, the variation of the angle between 31.5 to 9.1°, revealed good compatibility, an interesting fact since squalane and the natural counterpart squalene, have been reported to have relevant properties from the skin point of view, with antioxidant, detoxifying and regeneration activities, also acting as drug carrier in both in vivo models and in vitro environments" – better would be "the change of the angle from 31.5 to 9.1°…". Besides, this sentence is too big, too much commas which are not necessary, and make the sentence difficult to understand. The sentence could be divided into two or three sentences.

L. 275. "…that includes, the degree of the polymer hydrophilicity…" – comma use.

L. 282. "Additionally, to evaluate the capacity of the manufactured polyester to entrap water soluble molecules, was used the rhodamine B base (RHB) as a model" - Additionally, to evaluate the capacity of the manufactured polyester to entrap water soluble molecules rhodamine B base (RHB) was used as a model" – Comma use, word order.

L. 329. "…waxes are removed thorugh a crystallization process." - …through…

L. 353-354. "Epoxidation and hydroxylation reaction yields were based on the FDR initial input were determined according to equation (1) and expressed in percentage" – maybe "Epoxidation and hydroxylation reaction yields were based on the FDR initial input and were determined according to equation (1) "

L. 364. "… the cure was carried in a vacuum drying oven…" – should be "… curing was carried out in a vacuum drying oven…"

L. 416. "Since for biopolymers, is not expected an equilibrium in contact angle, but rather…" – in English the word order should be "Since for biopolymers no contact angle equilibrium is expected, but rather ..."

Author Response

First, at all, author’s would like to thank all the suggestions given by the reviewers under supervision of the editor, author’s have carefully undertaken the revision requests for your reconsideration. All the changes made in the manuscript are highlighted at blue color directly in text and the identified in the reviewer’s letter with the line number.

Comment 1This paper presents a study of the synthesis and properties of polyester based on biowaste, a microbial oil, and castor oil to produce a porous scaffold for wound dressing. The manuscript contains some interesting results, but it needs a major revision especially in the synthesis section.

Comment 2: First, I don't understand what is the composition of the FDR used in the synthesis. Authors claimed that it contains "hydrocarbons (43.29 g/100 g lipids), a small percentage of complex lipids such as tri- and di-glycerides along with farnesol and terpenes". These data are taken from Ref. 46. But authors also claimed that the winterization was applied to purify and provide some uniformization of this reagent. And what was the composition of FDR after winterization? What were the components for epoxidation: unsaturated hydrocarbons, farnesene, farnesol, glycerides?

Answer: The focus of this study was the valorization of a biowaste (FDR) derived from of a sugarcane fermentation process through the production of building blocks (e.g. epoxide and polyols) to the production of condensation polymers. This waste was considered suitable to be applied in the production of flexible polyesters for biomedical application. FDR is a very complex oily mixture similar to a vegetable oil, as it can be seen in the mentioned manuscript (https://www.mdpi.com/1424-8247/14/6/583). It has a relatively uniform composition between process batches, but it has an uncertain amount of suspended wax esters that crystallize around room temperature, which are quite difficult to remove by a simple filtration or centrifugation. The first strategy was to directly use the crude mixture as raw material; however, the reaction of epoxidation resulted in the formation a product highly viscous at room temperature. That characteristic is not appealing from the industrial and market point a view, hence the removal of waxes by crystallization (winterization) a process commonly used in vegetable oils (e.g. olive oil and sunflower oils). This simple technique proved to be an efficient way to remove those waxes, providing the expected uniformization and consequent reduction of viscosity of derived materials. Structurally is not visible a relevant difference in composition (in Figure 2, it was included FTIR spectra of FDR and FDRw). The winterized sample was then characterized, and the iodine value of 130 g I2/100 g reveals a certain degree of unsaturation, C=C bonds, which are reactive sites prone to be functionalized by epoxidation. Additional data was provided in the revised article on lines 84-112 and Figure 1.

Comment 3: Second, the question concerning the synthesis of polyol and the description of its IR-spectrum. 

Answer: Indeed the mechanism to the synthesis of polyol that was pointed out is incorrect since it wasn’t pure ricinoleic acid the ring-opening agent used (e.g. http://dx.doi.org/10.1021/acssuschemeng.5b00001). The effective role of castor oil is to increase the hydroxyl content of the mixture, since FDRw and the derived epoxide already contain hydroxyl groups. The FTIR analysis, now based on Figure 2, was corrected in the revised article on lines 160-170.

Comment 4: L.149-151. Authors wrote: "This is a result of the functionalization of the epoxide by the castor oil fatty acids, mainly ricinoleic acid which has specifically two functional groups, ester, and hydroxyl" – ricinoleic acid does not contain ester groups. It is an acid. If authors mean that castor oil (which contains up to 90% ricinoleic acid triglycerides) is a functionalizing agent, they should rewrite this sentence. And another question – how the authors imagine the functionalization of epoxide with castor oil. What is the reactant interacting with epoxy groups – triglycerides of ricinoleic acid (which contain hydroxyl groups in their structure) or free fatty acids also present in castor oil? But I did not observe the decrease in the intensity of the band corresponding to epoxide ring (831-863cm-1). And taking into account that the polyol synthesis was carried out simply by mixing epoxide with castor oil at elevated temperature without any separation or purification I do not see any strong evidence of the polyol synthesis.

Answer: This subject was already addressed in the answer of the previous comment (comment 3) and the correction was included in the revised article on lines 127 to 141 and lines 358 to 361 and lines 381, 383.

Comment 5. Authors wrote in the Conclusion that "…microbial oil … was used as feedstock in the synthesis of an unsaturated aliphatic and linear copolyester". How did authors synthesize linear copolyester via using polyol, azelaic diacid and cross-linker? The use of polyol (by the way, what is the average number of OH-groups per molecule of polyol?) should give branched product and cross-linker use results in the formation of polymer network.

Answer: in Line 119 of the original version “The final polyol showed a value of 5.5 hydroxyl groups per molecule”. The synthesis of a linear copolyester with similar composition was found in reference 10.1021/acssuschemeng.5b00001. However, soybean oil has by far a less complex lipid composition than the feedstock used in this study. In fact, with this functionality and without further analysis it’s not possible to conclude that the obtained polyester is linear. That observation was removed in the revised article.

Comment 6: So, the main problem in the synthesis section is the lack of understanding of which substances are the initial reagents and which are formed as a result of the synthesis. Therefore, the authors must adjust the synthesis methods and perform additional analyzes to be sure of the composition and structure of the initial and final product. Besides, general schemes of occurred reactions should be added to the text.

Answer: Indeed, is quite difficult to fully understand the profile complexity of this biowaste, A reactional scheme should be quite inaccurate. The focus of this article was the valorization through the synthesis of polymeric materials, in this case an epoxide to the synthesis of a biocompatible flexible polyester. From the industrial point a view, the chemical/physical characterization provided in the manuscript it was considered to be sufficient and because of that no further analysis was performed.

Other remarks:

Comment 7: L. 104. "…the iodine value, presented the expected sharp decrease after FDR epoxidation from 130 to 32 mg KOH.g-1…" Iodine value is expressed as the mass of iodine per 100g of substance, not mg KOH.

Answer: This was a typing mistake and authors have corrected that in the revised article, now on line 123-125, and Table 1.

Comment 8: L.116-118. – " The influence of reaction conditions was clearly demonstrated by the synthesis of different polyols based in soybean oil through oxirane cleavage with water catalyzed by perchloric acid, with -OH values of 160, 240, and 285 mg KOH.g-1 [52]." – I don't understand the sense of this sentence. If authors wanted to demonstrate the influence of the reaction conditions onto hydroxyl value, it should be pointed in the text what reaction conditions caused such change in -OH value. 

Answer: In the light of this revision that sentence was removed.

Comment 9: L.122 and Table 1. PDI value of polyol presented in this table and in the text is equal to 0.99. How can it be? PDI=(Mw/Mn). And weight average molecular weight is always larger than number-average. Should I believe to other data after that? Another question: what molecular weight is presented in the table: Mw or Mn? And I have never seen molecular weight determined to tenths of a Dalton via liquid chromatography.  

Answer: Size exclusion chromatography technique was used in several articles (e.g. https://pubs.acs.org/doi/full/10.1021/ma011341f. In fact, Mw and Mn values are quite similar. In the revised article is clarified the technique and the Mn value was included.

Comment 10: L.131-132. "The reduction to the final value (4.2 mg KOH.g-1) for the polyol, is an improvement of the hydrolysis resistance" – I don't understand what authors would like to say in this sentence. Please rephrase it. In my opinion, the decrease in acid number can be explained by the introduction in the system of castor oil with rather low acid value (0,91 mg KOH/g - doi:10.1021/acsomega.0c05810).

Answer: Indeed that citation is misplaced, and it was removed. In fact, it’s the presence of castor oil that is considered to be connected to the hydrolysis resistance of the final polymers (e.g. https://www.nature.com/articles/pj201611).

Comment 11: L. 147-148. "In the polyol screening, the peaks exhibited at 2930 cm−1 and 2855 cm−1 assigned to alkane groups are better defined and are visible in the increasing intensity of the bands at 3462 cm-1 and 1734 cm-1, assigned to -OH and C= O stretching, respectively". I don't understand "alkane groups are better defined" – better than where or better than what? And how peaks assigned to alkane groups " are visible in the increasing intensity of the bands at 3462 cm-1 and 1734 cm-1"? Is it simply poor English or something else?

Answer: Redaction and analysis was improved in the revised article, now on lines 159-170.

Comment 12: L.189. "…a Young elastic modulus between 1.9×10-3 and 2.2×10-3 MPa." – The data of Table 2 witness that Young elastic modulus is between 1.6×10-3 and 2.2×10-3 MPa.

Answer: The information in the table was corrected now in the revised article (Table 1).

Comment 13: L. 325. "Castor oil (non-edible oil) with 85-90 % content in ricinoleic acid" – Castor oil does not contain 85-90% ricinoleic acid. It contains 85-90% triglycerides of ricinoleic acid. Authors should determine acid value and maybe hydroxyl value of castor oil to be sure of the free acid content in the oil composition. Because the acid number for castor oil ranges usually from 0.14 to 1.97 mgKOH/g oil (doi:10.1590/fst.19620), while acid number for ricinoleic acid is 175-190 mg KOH/g.

Answer: The characterization provided from the castor oil supplier was included in the revised article on lines 358-361.  

Comment 14: L. 353-354. "Epoxidation and hydroxylation reaction yields were based on the FDR initial input were determined according to equation (1) and expressed in percentage: Y=(mf/mi)x100, where, mf is the epoxide or polyol mass, and mi is …FDR initial mass". – I don't understand how the polyol mass was determined if after the hydroxylation reaction "…no further purification was necessary." (L. 352). In such case the yield will be higher than 100%. How did authors separate final product from castor oil?

Answer: Indeed it’s incorrect. The value pointed out it’s not the polyol yield but rather the polyester yield based on the FDRw-based polyol initial input. It was determined in real laboratory conditions by gravimetry. The formula definition is corrected in the revised article (lines 406 to 409), value eliminated from Table 1 and added in lines 158-159.

Comment 15: L. 421. Gel-content determination. Did the authors grind or cut the samples to determine the gel-fraction? What temperature was used for extraction?

Answer: Details were included in the revised article on lines 471-476.

Comment 16: L. 438. Equation for water absorption calculations. %=(Wt-Wd)/Wt. Water absorption is calculated as the difference between the weight of soaked sample and dry initial sample divided by weight of initial weight of dry sample. So, the equation 5 should be as follow: %=(Wt-Wd)/Wd

 Answer: Corrected in the revised article (Line 486).

Comment 17: Some references are out of order. For example:

P.5, L. 167. (Menzies and Jones, 2010). – I didn't find this reference in the References section.

P.9, L. 341. "Epoxidation (Epoxide) was performed according to [100] with slight modifications" – Reference [100] does not contain any technique for epoxidation. Please double check this reference.

Answer: A mistake. Same author, same year, different article. Corrected.

P.13., L. 514. (Korsmeyer et al., 1983) – I didn't find this reference in the References section.

Answer: References were all corrected in the revised article.

Comment 18: English also should be improved.

There are many mistakes (such as word order, comma use, wrong words etc.) that make it difficult to understand the tex. See some of them below please.

L 73-74. "Several natural synthetic polymers have been used for wound healing applications since they join properties, such as biocompatibility and/or biodegradability…" – "…they combine properties…" sounds better. Correction in line 73-76.

L. 76-77. "…since to achieve the required properties for tissue regeneration, biopolymers, have high production costs." – Please rephrase this sentence. Phrase reformulated in lines 73- 76

L. 117. "…based in soybean oil…" – in English should be "…based on soybean oil…". Corrected in line 117.

L.154. "The contact angle value stabilized within the time frame of 60 s, for both fluids, and for water, showed (Figure 3) a variation between 70.2° and 54.9°" – Please double check comma use. Corrected in lines 182-186.

L. 167-171. "Regarding the squalane oil, the variation of the angle between 31.5 to 9.1°, revealed good compatibility, an interesting fact since squalane and the natural counterpart squalene, have been reported to have relevant properties from the skin point of view, with antioxidant, detoxifying and regeneration activities, also acting as drug carrier in both in vivo models and in vitro environments" – better would be "the change of the angle from 31.5 to 9.1°…". Besides, this sentence is too big, too much commas which are not necessary, and make the sentence difficult to understand. The sentence could be divided into two or three sentences. Corrected in lines 185-189.

L. 275. "…that includes, the degree of the polymer hydrophilicity…" – comma use.

L. 282. "Additionally, to evaluate the capacity of the manufactured polyester to entrap water soluble molecules, was used the rhodamine B base (RHB) as a model" - Additionally, to evaluate the capacity of the manufactured polyester to entrap water soluble molecules rhodamine B base (RHB) was used as a model" – Comma use, word order. Correction in lines 302-304.

L.329. "…waxes are removed thorugh a crystallization process." - …through… Correction in line 358.

L.353-354. "Epoxidation and hydroxylation reaction yields were based on the FDR initial input were determined according to equation (1) and expressed in percentage" – maybe "Epoxidation and hydroxylation reaction yields were based on the FDR initial input and were determined according to equation (1) " Correction in lines 414-415.

L.364. "… the cure was carried in a vacuum drying oven…" – should be "… curing was carried out in a vacuum drying oven…" Corrected in line 391.

L. 416. "Since for biopolymers, is not expected an equilibrium in contact angle, but rather…" – in English the word order should be "Since for biopolymers no contact angle equilibrium is e

Round 2

Reviewer 2 Report

I have read the revised version of the article and author’s responses to my comments. Most of the authors' answers satisfied me. However, some errors need to be corrected before the article is published.

1. The first question concerns molecular weight and polydispersity index. Authors wrote that they used β-Farnesene distillation oily residue as the initial product for the synthesis of polyester. This biowaste is a mixture of different substances – hydrocarbons, terpenes, lipids. It was firstly epoxidized and then mixed with castor oil which, in turn, is also not an individual substance. How, in this case, the polydispersity index of obtained product is equal to 1,0? I can't believe this. Did the authors determine the molecular weight and polydispersity index of initial or winterized FDR? And what is the reason to determine molecular weight for only this product? I would recommend the authors to remove these data from the table and text, as they do not carry any important information for the article.

2. L. 89, 99, 108. Authors claimed that they carried out the winterization in order to remove waxes esters and decrease in such a way  the product viscosity (L. 89 and 99), and a few rows later – “FDRw (FDR winterized) was then submitted to a typical epoxidation to increase the oil viscosity along with thermal and oxidative stability therefore improving the lubricant [45] and plasticizing properties” (L.108). what is the point of first reducing the system viscosity, and then increasing it. Maybe it makes sense to remove this part of the sentence (about the increase of viscosity) and keep only the second part: “…to increase thermal and oxidative stability therefore improving the lubricant [45]and plasticizing properties.

3. Table 2 and L. 208. I wrote in my first review about the discrepancy between the Young's modulus values in the text and in the table. But the correction made by the authors only worsened the situation. Please check the data in Table 2. Young's modulus is 1.9×10-3±2.2×10-3. That means that the deviation is greater than the determined value!!!

4. L. 396. “… particle size between 315 and 500 mm” – 500mm???, maybe mm or nm?

5. L.443. “Polyol molecular weight distribution was determined by size-exclusion chromatography using high performance liquid chromatography” – What does it mean? Size-exclusion chromatography is one of the types of high performance liquid chromatography. Maybe, authors mean high performance liquid chromatograph?

English language use.

I recommend authors to carefully check the English language in the article. There are a lot of errors. Some of them but not all are listed below:

L.115 “All the products were characterized by acid and hydroxyl values as well viscosity, and density” –  should be “All the products were characterized by acid and hydroxyl values as well as viscosity, and density

L.124. “…insaturation bonds” – “unsaturated bonds

L.131. “After addition of castor oil (ratio 3:1) achieved a value of 104.2 mg KOH.g-1” - “After addition of castor oil (ratio 3:1) a value of 104.2 mg KOH.g-1 was achieved

L.394. “…a easily available and inexpensive technique…” – article “an” should be used - “…an easily available and inexpensive technique…”  

Author Response

First, at all, we would like to thank all the suggestions given by the reviewer under supervision of the editor, author’s have carefully undertaken the revision requests for your reconsideration. All the changes made in the manuscript are highlighted at blue color directly in text and the identified in the reviewer’s letter with the line number.

Reviewer

I have read the revised version of the article and author’s responses to my comments. Most of the authors' answers satisfied me. However, some errors need to be corrected before the article is published.

Comment 1. The first question concerns molecular weight and polydispersity index. Authors wrote that they used β-Farnesene distillation oily residue as the initial product for the synthesis of polyester. This biowaste is a mixture of different substances – hydrocarbons, terpenes, lipids. It was firstly epoxidized and then mixed with castor oil which, in turn, is also not an individual substance. How, in this case, the polydispersity index of obtained product is equal to 1,0? I can't believe this. Did the authors determine the molecular weight and polydispersity index of initial or winterized FDR? And what is the reason to determine molecular weight for only this product? I would recommend the authors to remove these data from the table and text, as they do not carry any important information for the article.

Answer: Those parameters were only considered relevant for the formulated polyol, since molecular weight, and functionality, are technical properties important for polymer cross-linking density, conditioning final thermal and mechanical properties. Since Mn and polydispersity (a mathematical ratio) are parameters not considered to be of importance for this purpose, that information was removed as suggested by the reviewer.

Comment 2. L. 89, 99, 108. Authors claimed that they carried out the winterization in order to remove waxes esters and decrease in such a way  the product viscosity (L. 89 and 99), and a few rows later – “FDRw (FDR winterized) was then submitted to a typical epoxidation to increase the oil viscosity along with thermal and oxidative stability therefore improving the lubricant [45] and plasticizing properties” (L.108). what is the point of first reducing the system viscosity, and then increasing it. Maybe it makes sense to remove this part of the sentence (about the increase of viscosity) and keep only the second part: “…to increase thermal and oxidative stability therefore improving the lubricant [45] and plasticizing properties.

Answer: The phrase was not correctly formulated, and therefore that inference was removed and the sentence rephrased as can be read now on lines 108-109. The intention of the oil epoxidation is not to increase viscosity, but rather a common effect. Quote from http://dx.doi.org/10.1016/j.jiec.2016.02.008 “Epoxidation of vegetable oils generally results in increased oxidative stability, better acidity value, increased adsorption to metal surfaces which results in better lubricity, increased viscosity, decreased viscosity index, and increased pour point.

In fact, FDR winterization presented a great impact in the viscosity reduction of the derived products, namely of the epoxide, when comparing with the epoxide derived from crude FDR. However that doesn’t mean that the epoxide is less viscous than the feedstock.

Comment 3. Table 2 and L. 208. I wrote in my first review about the discrepancy between the Young's modulus values in the text and in the table. But the correction made by the authors only worsened the situation. Please check the data in Table 2. Young's modulus is 1.9×10-3±2.2×10-3. That means that the deviation is greater than the determined value!!!

Answer: Author’s would like to thank’s reviewer attention. This was a mistake that was not properly corrected. This was revised and changed in Table 2 and line 205.

Comment 4. L. 396. “… particle size between 315 and 500 mm” – 500mm???, maybe mm or nm?

Answer: Font error. The correct units are mm. Corrected in lines 283 and 392.

Comment 5. L.443. “Polyol molecular weight distribution was determined by size-exclusion chromatography using high performance liquid chromatography” – What does it mean? Size-exclusion chromatography is one of the types of high performance liquid chromatography. Maybe, authors mean high performance liquid chromatograph?

Answer: Correction in line 439.

Comment 6. English language use.

I recommend authors to carefully check the English language in the article. There are a lot of errors. Some of them but not all are listed below:

L.115 “All the products were characterized by acid and hydroxyl values as well viscosity, and density” –  should be “All the products were characterized by acid and hydroxyl values as well as viscosity, and density

L.124. “…insaturation bonds” – “unsaturated bonds”

L.131. “After addition of castor oil (ratio 3:1) achieved a value of 104.2 mg KOH.g-1” - “After addition of castor oil (ratio 3:1) a value of 104.2 mg KOH.g-1 was achieved

L.394. “…easily available and inexpensive technique…” – article “an” should be used - “…an easily available and inexpensivHoe technique…”  

Answer: The revised article was fully edited by an English native speaker, eliminating obvious errors and reformulating entire sentences. The most important changes are marked in blue.